# STOP-Colitis pilot trial protocol: a prospective, open-label, randomised pilot study to assess two possible routes of faecal microbiota transplant delivery in patients with ulcerative colitis

Mohammed Nabil Quraishi  ,[1,2] Mehmet Yalchin,[3] Clare Blackwell,[4] Jonathan Segal,[3] Naveen Sharma,[5] Peter Hawkey,[6] Victoria McCune,[6,7] Ailsa L Hart,[3] Daniel Gaya,[4] Natalie J Ives,[8] Laura Magill,[8] Shrushma Loi,[8] Catherine Hewitt,[8] Konstantinos Gerasimidis,[9] Nicholas James Loman  ,[6] Richard Hansen,[10] Christel McMullan  ,[11] Jonathan Mathers,[12] Christopher Quince,[13] Nicola Crees,[14] Tariq Iqbal[1,15]

**Correspondence to**
Professor Tariq Iqbal;
t.h.iqbal@bham.ac.uk

## ABSTRACT

**Introduction** Imbalance of the gut microbiome is key to the pathogenesis of ulcerative colitis (UC). Faecal microbiota transplant (FMT) is the transfer of homogenised and filtered faeces from a healthy individual to the gastrointestinal tract of a patient with disease. Published datasets show a positive signal for the use of FMT to treat UC, but the optimal route and dose of FMT remain unanswered.

**Methods and analysis** This prospective, multi-centre open-label, randomised pilot study will assess two possible routes of FMT delivery, via the nasogastric (NG) route or by delivery to the COLON, in 30 patients with active UC recruited from three sites in the UK. Stool will be collected from healthy screened donors, processed, frozen and stored under a Medicines and Healthcare products Regulatory Agency (MHRA) "specials" manufacturing licence held at the University of Birmingham Microbiome Treatment Centre. Thawed FMT samples will be administered to patients either via eight nasogastric infusions given initially over 4 days starting on the day of randomisation, and then again for 4 days in week 4 for foregut delivery (total of 240 g of stool) or via one colonoscopic infusion followed by seven weekly enemas according to the hindgut protocol (total of 360 g of stool). Patients will be followed up weekly for 8 weeks, and then at 12 weeks. The aims of this pilot study are (1) to determine which FMT administration route (NG or COLON) should be investigated in a randomised double-blind, placebo-controlled trial and (2) to determine if a full randomised controlled trial is feasible. The primary outcome will be a composite assessment of both qualitative and quantitative data based on efficacy (clinical response), acceptability and safety. At the end of the pilot study, decisions will be made regarding the feasibility of a full randomised double-blind, placebo-controlled trial and, if deemed feasible, which route of administration should be used in such a study.

## Strengths and limitations of this study

► Matching of single donor with each patient—this will allow detailed exploration of donor characteristics associated with a favourable patient response.

► Intensive faecal microbiota transplant (FMT) treatment regimen—shown to be a major contributing factor in clinical response to FMT.

► Assessment of mechanistic outcomes and donor diet—this will help in understanding the mechanisms associated with response to FMT.

► Qualitative data from clinicians and patients informing the potential future randomised controlled trial (RCT).

► This study is limited to assessing the feasibility of an RCT and the best route of FMT delivery in ulcerative colitis (UC) patients. The assessment of the efficacy of FMT in achieving and maintaining remission for patients with UC needs to be tested in a future full-scale RCT.

**Ethics and dissemination** Ethical approval for this study has been obtained from the East Midlands-Nottingham Research Ethics Committee (REC 17/EM/0274). At the end of the study, findings will be reported at national and international gastroenterology meetings and published in peer-reviewed journals.

**Trial registration number** ISRCTN74072945

## INTRODUCTION

In the inflammatory bowel diseases (IBDs), Crohn's disease (CD) and ulcerative colitis (UC), there is imbalance or 'dysbiosis' of the gut microbiota compared with the healthy bowel, with individuals having IBD showing reduced bacterial diversity compared with

healthy individuals.[1][2] Indeed, the currently accepted paradigm concerning the pathogenesis of IBD involves an aberrant immunological response to the intestinal microbiota in a genetically susceptible host.[3] Whether the observed dysbiosis represents 'cause' or 'effect' remains unanswered. Faecal microbiota transplant (FMT) is the infusion of a faecal suspension from a healthy donor into the gastrointestinal tract of a patient with disease, and there is interest in the potential of this technique for modifying the gut microbiome as a potential treatment for IBD.[4]

Since the first descriptions of CD and UC at the beginning of the 20th century, it has been strongly suspected that the gut microbiota may have a defining role in the pathogenesis of IBD.[5] Early culture-based studies underestimated the complexity of the microbiota, and were hampered by various culture biases and the challenge of enumerating fastidious bacteria with complex growth conditions. With the advent of cheap high-throughput genetic sequencing techniques allied with complex bioinformatics capability, there has been a revolution in our understanding of the composition and function of the colonic microbiome. As a result of studies on the microbiome both in patients with IBD and animal models, we know that patients with IBD (either CD or UC) have, at the phylum level, a reduction in Firmicutes and a relative increase in Proteobacteria.[2][6]

Accumulating data suggest that alteration in the gut microbiome plays a central role in driving UC; datasets highlighting the importance of *Roseburia hominis*,[7] *Faecalibacterium prausnitzii*[8] and *Akkermansia muciniphila*[9] mediating anti-inflammatory responses in UC have been published. Attempts to alter the microbiome with probiotics, while disappointing in CD, have shown some promise in UC.[10][11] Since the first exploratory use of FMT to treat UC in 1989,[12] numerous case series have been published demonstrating encouraging efficacy signals.[13] This has led to investigators testing FMT as a treatment for UC and to date there have been four randomised controlled trials (RCTs) of FMT for the treatment of UC.[14] The first was a trial of fresh and frozen enemas which were delivered weekly for 6 weeks, water being used as placebo.[15] There was a significant increase in remission at the primary endpoint in the active versus placebo arm. In the second study, fresh FMT by nasojejunal delivery was used and autologous stool was used as placebo.[16] Although the results favoured active treatment, they were not statistically significant. The last two RCTs were reported in 2017[17][18] and both used frozen FMT from pooled donors and involved intensive treatment regimens. Both achieved statistically significant results in favour of active treatment, with remission rates comparing active and placebo of 32% vs 9%[18] and 27% vs 8%, respectively.[17]

As a result of this recent work, there is great interest in FMT as a possible treatment for UC, but the optimal route of delivery remains unknown. Before we can undertake a full-scale double-blind RCT of FMT versus placebo, we need to undertake a pilot study to determine the optimum route of FMT delivery and assess the feasibility (recruitment, protocol adherence) of undertaking a trial of FMT in UC. In this pilot study, we will compare nasogastric (NG) versus colonoscopic (COLON) delivery of FMT, using an intensive treatment regimen, in patients with active UC to investigate the optimum route of FMT delivery for the treatment of UC. We will assess the effectiveness and acceptability of the two routes of FMT delivery, and whether FMT by either NG or COLON is suitable to take forward to a full RCT against placebo, and whether such a trial is feasible. Here we describe the protocol for the STOP-Colitis pilot study.

## METHODS AND ANALYSIS
### Study design
A prospective, multi-centre open-label, randomised pilot study to assess two possible routes of FMT delivery for the treatment of UC. Thirty patients with active UC will be randomised to receive FMT via either the NG route or by direct delivery to the COLON. Patients will be recruited from three hospitals in the UK (Queen Elizabeth Hospital, Birmingham; St Mark's Hospital, London and Glasgow Royal Infirmary).

At the end of the pilot study, the data will be reviewed by an Independent Oversight Committee (IOC), consisting of two consultant gastroenterologists and a professor of medical statistics and clinical trials, who, according to the prespecified STOP/GO criteria (see Analysis of outcome measures below) will decide (1) which route of FMT administration (NG or COLON) is most appropriate to investigate in the full randomised double-blind, placebo-controlled trial and (2) whether it is feasible to proceed to the main RCT.

### Sample size
As this is a pilot study, no formal sample size calculation was undertaken as the study is not designed or powered to detect a statistically significant difference in efficacy between the two FMT methods of delivery. The recruitment target for this study is 30 patients.

### Aims and objectives
The aims of this pilot study are as follows:
1. To determine which FMT administration route (NG or COLON) should be investigated in a randomised double-blind, placebo-controlled trial.
2. To determine whether a full RCT is feasible.

In order to achieve these aims, the pilot study has the following clinical (quantitative), qualitative and mechanistic objectives.

### Clinical objectives
To assess:
1. Whether FMT by the NG route induces clinical response in patients with active UC.
2. Whether FMT by the COLON route induces clinical response in patients with active UC.
3. Tolerability and safety.
4. Which route of FMT delivery (if any) is suitable to investigate in a RCT.

## Qualitative objectives

To assess:

1. Patient acceptability of FMT (NG).
2. Patient acceptability of FMT (COLON).

## Mechanistic objectives

To assess:

1. Whether FMT by either route is associated with a change in faecal calprotectin as a surrogate marker of colonic inflammation.
2. Changes in the colonic microbiome and metabolome (short chain fatty acids (SCFA)) induced by FMT via each route.
3. Reduction in C-reactive protein (CRP).

## Other objectives

1. Effect of diet (donors) on microbiome.
2. Effect of time from stool donation to treatment on success of FMT.

## Outcome measures

The primary outcome will be a composite assessment of both qualitative and quantitative data relating to efficacy, acceptability, safety and mechanistic outcomes.

## Clinical outcome measures

1. Clinical response (primary measure of efficacy) defined as ≥3 point reduction in the full Mayo score from randomisation to week 8, and 30% reduction from randomisation and at least one point reduction of rectal bleeding sub-score or an absolute rectal bleeding sub-score of 0 or 1.
2. Time to clinical response (where clinical response is defined as ≥2 point reduction in partial Mayo).
3. Clinical remission at week 8 (full Mayo score of ≤2, with no subscore >1).
4. Participant's weight at weeks 8 and 12.
5. Quality of Life (QoL) using the generic Short-Form 36 (SF-36) and the disease specific Inflammatory Bowel Disease Questionnaire (IBDQ) at weeks 8 and 12.
6. Adherence to FMT.
7. Adverse events (AE) and serious adverse events (SAE).

## Qualitative data

Patient acceptability will be assessed through qualitative research interviews. During these interviews, patients from both groups will be asked about their perspectives and experience of FMT during the pilot trial, including receiving the intervention, recovering from the intervention and their symptoms and QoL. Further examples of interview questions can be found in the online supplementary file 1.

## Mechanistic outcome measures

1. Faecal calprotectin.
2. Measures of microbiome (faecal and mucosal)—specifically shifts in alpha diversity following FMT.
3. Mucosal healing.
4. Urinary metabolome (SCFA).
5. CRP.

## Other outcome measures

1. Association between the donors' and recipients' dietary profile and microbiome.
2. Time, that is, number of days, from donor stool processing to treatment for association between efficacy and freezer life of FMT.

## Donor sample acquisition and processing

Donors will be recruited following advertisement from healthy unrelated anonymous individuals living in Birmingham. We are excluding healthcare workers due to their potential exposure to microbes affecting the microbiome. Donors must be ≥18 and<50 years of age, have a normal morning bowel habit, have a normal body mass index (≥18.5 and ≤25), be non-smokers (not smoked for at least 12 months) and have no recent history of diarrhoea or rectal bleeding. Potential donors will undergo rigorous screening using a health-screening questionnaire. Those who are eligible following this screening will be consented for the donation process and have blood and faecal samples taken to test for transmissible pathogens in accordance with UK, American Gastroenterology Association (AGA) and European guidelines.[19–21] Individuals who pass the screening process will be invited to donate morning faecal samples for 10 days and to deliver these for processing within 6 hours of defaecation. At first donation, they will also be asked to complete the EPIC-Norfolk food frequency questionnaire.[22]

Donations will be processed and stored under an MHRA manufacturing licence at the University of Birmingham Microbiome Treatment Centre. Stool collection and processing were performed under aerobic conditions. Donations will be prepared in saline with 10% glycerol added as a cytoprotectant. Samples will then be stored frozen at −80°C for up to 24 weeks. Samples will be dispatched for use at each site as required. An aliquot from each donated sample will be kept for analysis in the event of AEs occurring as a result of FMT and samples from each donation period will be sent for genetic sequencing and metabolic analysis. At the end of each 10-day donation period, donors will receive financial compensation and will undergo an exit health questionnaire and be asked to provide a further stool sample to test for pathogens. Further information on the donor sample processing will be present in the protocol and a laboratory standard operating procedure (SOP) for donor preparation (available on request).

## Participants

### Inclusion criteria

1. Adult patients (aged between 16 and 70 years) with clinically confirmed UC for at least 12 weeks prior to the screening visit.
2. Partial Mayo score of ≥4 and≤8 despite stable disease maintenance treatment with 5-aminosalicylates with or without immunomodulators, or on no treatment.
3. Rectal bleeding subscore of ≥1 on the partial Mayo;.
4. Able to give written, signed informed consent.

### Exclusion criteria

1. Stool positive for *Clostridium difficile* or infection by either PCR or ELISA.
2. Positive for Hepatitis A/B/C and/or HIV infection.
3. Antibiotics in the preceding 12 weeks prior to date of the screening visit.
4. Systemic/topical steroids in the preceding 2 weeks prior to the date of the screening visit.
5. Biologics in the preceding 12 weeks prior to the date of screening visit.
6. Commercial probiotics and prebiotics in the preceding 12 weeks prior to the date of the screening visit.
7. On oral nutritional supplements or enteral/parenteral nutrition in the preceding 4 weeks prior to the date of the screening visit.
8. Pregnant or lactating.
9. Not willing to take appropriate contraceptive measures to prevent pregnancy during trial participation.

### Participant enrolment

Potentially eligible patients who express an interest in participating in the trial will be consented via a two-stage consent process. Participant information sheets will be provided to facilitate the consent process.

### Registration and screening visit

The first stage will involve consent for trial-specific screening activities, and consent to collect stool and urine samples for the mechanistic substudies. Patients will undergo basic physiological assessments (pulse, blood pressure, temperature, height and weight) and baseline blood tests. They will be provided with a diary to record bowel symptoms (so that the partial Mayo score can be calculated at the randomisation visit), stool sample kits and bowel preparation kits (Moviprep). They will be asked to return the *C. difficile* stool sample as soon as possible, so that the result is available prior to the randomisation visit.

Following the screening visit, the qualitative researcher will arrange for an interview with the patient to take place prior to their randomisation visit (see Randomisation visit below). The research team at the hospital will contact the patient to notify them of their stool result. If they tested negative for *C. difficile*, they will be invited to attend the randomisation visit. Instructions will be given on when to take the bowel preparation, and they will be asked to collect a stool sample on the same day prior to taking the bowel preparation, which they will bring to the randomisation visit.

### Randomisation visit

The second stage is consent for entry into the trial. Following confirmation of all eligibility criteria and consent to randomisation, basic physiological assessments will be undertaken, blood test results will be checked, a urine sample will be taken for pregnancy testing in women and metabolomics and the partial Mayo score will be calculated from diaries. The patient will be asked to complete the baseline QoL questionnaires—IBDQ

and SF-36. All patients will have a colonoscopy to assess disease (done after randomisation), so that a full Mayo score can be calculated, and to collect mucosal biopsies.

### Randomisation

Patients will be randomised at the level of the individual in a 1:1 ratio to either NG or COLON delivery of FMT. Randomisation will be provided by a computer-generated program at the Birmingham Clinical Trials Unit. A minimisation algorithm will be used to ensure balance in the treatment allocation over the following variables:

1. Partial Mayo score (4–5 or 6–8).
2. Current smoking status (current smoker: yes or no (not smoked for the past 12 months)).

A 'random element' will be included in the minimisation algorithm, so that each patient has a probability (unspecified here), of being randomised to the opposite treatment that they would have otherwise received.

### Interventions

Before the initial FMT treatment, all patients will receive standard bowel preparation: 2 L of reconstituted Moviprep solution within the 24 hours prior to procedure.

### NG treatment

Patients randomised to NG treatment will be pretreated with a proton pump inhibitor (lansoprazole 30 mg) and a prokinetic agent (domperidone 10 mg) at least 30 min prior to each FMT infusion to reduce gastric acid secretion and prevent the risk of gastro-oesophageal regurgitation. Naso-gastric tubes (NGTs) will be positioned and checked for correct position as per local protocol. Following thawing at room temperature, 50 mL thawed FMT containing 30 g faeces will be infused. Following the first treatment, patients will return for the next 3 days for further treatment following an overnight fast. It will be the individual's choice whether they wish to retain the NGT for the 4-day treatment course or have the tube removed and repassed every day. On the last day of treatment, the NGT will be removed to be replaced for the second course of treatment. At week 4, patients will return for a further four FMT infusions over 4 consecutive days following a fast from midnight.

In summary, patients will receive 30 g FMT in 50 mL aliquots for NG administration each day for 4 days at the start of the trial (starting on the day of randomisation), and then again for 4 days in week 4 (total FMT dose 240 g).

### COLON treatment

Patients randomised to COLON treatment will receive a thawed 250 mL aliquot of FMT (containing 150 g faeces) on the day of randomisation. This will be delivered to the colon using a spray catheter, with 125 mL sprayed into the caecum to treat the right side of the bowel and the remaining 125 mL sprayed directly onto the rest of the colon. Following treatment, patients will receive a single dose of loperamide. Patients will return on a weekly basis to receive 100 mL enema up to week 7 containing 50 mL

of saline and 30 g faeces, followed by loperamide each time.

In summary, patients will receive 150 g of stool in 250 mL for colonic administration on the day of randomisation followed by 30 g of stool in 100 mL in normal saline for administration by enema for 7 weeks (total FMT dose 360 g).

## Data collection

All patients will be followed weekly up to week 8, and then again at week 12. At each visit, the partial Mayo score will be calculated from the patient diaries, and medication use and any adverse events occurring will be recorded. Stool samples will be collected at weeks 2, 4, 6, 8 and 12, and blood samples will be taken for CRP assessment at weeks 4, 6 and 8. At weeks 8 and 12, physiological data, a urine sample and the SF-36 and IBDQ QoL questionnaires will also be collected. At week 8, a flexible sigmoidoscopy will be undertaken to calculate the full Mayo score (see online supplementary figure 1).

## Timelines

We completed recruitment into this pilot study in April 2019. Data collection will be completed by July 2019 and analysis of the data will commence in August 2019.

## Qualitative research: patient and clinician experience and acceptability of FMT and trial processes

A comparison of the acceptability of the two routes of FMT being used in this study is key to the outcome of this pilot study. A qualitative researcher will conduct interviews with patients participating in both arms of the trial. Each patient will take part in a semistructured interview at two time points; following the screening visit prior to randomisation and after the 12-week follow-up visit. These one-to-one interviews will take place either in person, by telephone or Skype, as preferred by the patients. The initial interview will focus on participants' experience of UC and treatment to date, along with their understanding and expectations for FMT and the pilot study. A background questionnaire collecting sociodemographic and clinical details will also be administered. During the follow-up interview, patients' actual experience within the pilot study, including that of FMT (by NG and COLON) and of related trial processes and procedures will be explored. Interviews will also gather data on patient perspectives regarding important impacts and outcomes of FMT, thus providing the opportunity to compare these with the outcome measures selected for inclusion in the pilot and (later) full RCT. Further interviews with patients who withdraw from FMT treatment early and with a small sample of patients who decline consent to participate following screening (should this happen), will provide in-depth understanding of these decisions. Interviews will also be conducted with staff at the pilot sites to review their experience of trial conduct, for example, the acceptability and logistics of complex trial procedures within clinical environments. Interviews schedules can be found in online supplementary file 1.

Data from this qualitative research will contribute to an assessment of patient and clinician experience and acceptability of FMT within the pilot context, and along with recruitment and dropout rates, and data regarding clinical response, this will inform a decision about whether the NG or COLON route of delivery is taken forward to the full RCT. The qualitative data will also contribute to the refinement and optimisation of trial processes prior to the full RCT.

### Analysis of qualitative data

Interviews will be recorded with the consent of participants and transcribed clean verbatim for analysis. Analysis will be conducted with reference to recordings, transcripts and field notes taken at the time of data collection. A thematic analysis of content will be informed by the framework analytical approach.[23] Following initial familiarisation with the interview data, development of thematic frameworks and data coding will proceed in an iterative manner. Data collection and analysis will run concurrently so that emergent analytical themes can inform further data collection, and in particular comparative analytical questioning between patients allocated to NG or COLON.

## Mechanistic studies

There will be two aspects to mechanistic studies undertaken as part of this trial; non-invasive assessment of UC activity and assessment of patient's microbiome response to FMT in comparison with that of donor microbiome at baseline. The study will also investigate donor diet with regard to baseline microbiome and treatment effect.

### Non-invasive assessment of UC activity

This will be undertaken using the serum biomarker CRP and faecal calprotectin which will be measured using a standard protocol at one central laboratory in Glasgow.

### Microbiome-related assays

### 16S ribosomal DNA sequencing and metagenomics

16S ribosomal DNA sequencing will be performed on all samples collected during the study: donor faeces (fresh and frozen from day 1 and day 10 donations), faeces from the recipient at baseline, weeks 2, 4, 6, 8, and 12 and mucosal biopsies taken from the right colon, left colon and rectum at baseline and from left colon and rectum at week 8. Together these samples will be compared with clinical data, and correlations found with faecal calprotectin, in order to find the samples most likely to be biologically informative as markers of disease remission. Additionally, and based on the results of the initial 16S analysis, a shotgun metagenomics approach will be undertaken on around 200 selected samples to give insights regarding functional enrichment. For a certain subset of samples where additional strain resolution is required (eg, recovery of complete genomes of interest)

or linkage between extra-chromosomal elements such as plasmids and phages, further samples will be subject to long-read sequencing.

### Major bacterial metabolites and metabolomics on faecal and urine samples

Major bacterial metabolites including faecal short and medium chain fatty acids previously implicated in the aetiology and mucosal inflammation of IBD will be measured using gas chromatography and other assays as described previously.[24–26] Analysis of SCFAs in the samples will be carried out on faecal samples from the donors on day 1 and day 10 and on patients at baseline, weeks 8 and 12.

### Donor assessment of dietary intake

The habitual dietary pattern of the donors will be assessed using the validated food frequency questionnaire as used in the EPIC-Norfolk study in the UK.[27] Data from the EPIC questionnaires will be transferred to Glasgow for analysis. Energy, macronutrient and fibre intake will be estimated and expressed in nutrient ranks and quartiles and will also be compared against Department of Health recommendations and UK National Diet and Nutrition Survey results.

### Analysis of outcome measures

The primary outcome from the pilot study will be a recommendation about which route of administration (NG or COLON) to take forward to the main RCT, and whether a full RCT is feasible. The decision will be based on assessing the two methods of delivery using a composite assessment of both qualitative and quantitative data on efficacy using clinical response, acceptability and safety.

The primary comparison will comprise of those randomised to FMT by the NG route versus those randomised to FMT by the COLON route. All analyses will be based on the intention to treat principle. Data will be presented as summary statistics and differences between groups will be presented with 95% CIs from two-sided tests. No formal hypothesis testing will be undertaken and no p values will be presented.

### Primary outcome measures of efficacy for STOP/GO

The primary measure of efficacy for the STOP/GO is clinical response at week 8. The number and percentage of patients achieving a clinical response in the two treatment groups will be reported along with the 95% CI. A log-binomial model will be fitted to obtain a relative risk and 95% CI, adjusting for the minimisation variable smoking status and baseline full Mayo score.

### Secondary clinical outcome measures for STOP/GO

The number and percentage of patients achieving clinical remission in the two treatment groups will be reported. A log-binomial model will be fitted to obtain a relative risk and 95% CI, adjusting for the minimisation variables: smoking status and baseline partial Mayo score. Time to clinical response (based on the partial Mayo score) will be presented graphically as a Kaplan-Meier plot. A Cox

proportional hazards regression model will be fitted to obtain an HR and 95% CI. Continuous data (eg, weight, QoL scores) will be summarised using means and SD presented at each time point (baseline, weeks 8 and 12). The week 8 and week 12 data will be compared by treatment group using linear regression models with the minimisation variables and baseline values included in the model, with the mean differences and 95% CI presented.

Tolerability will be assessed quantitatively using adherence. Patients will be considered adherent if they receive at least 70% of their intended FMT dose. The number and percentage of patients who are deemed adherent to the FMT treatment in the two treatment groups will be reported along with the 95% CIs.

AE and SAE data will be tabulated. For each treatment group, the number of AEs and the number and percentage of patients experiencing an AE will be reported. Data on SAEs will be reported in the same way.

### STOP/GO guidelines

At the end of the pilot, the IOC will be convened to review the data. This group will make a recommendation to the Trial Management Group on (1) which route (if any) of FMT is appropriate to take forward to the full trial and (2) whether it is feasible to proceed to the full RCT.

To determine whether we will proceed forward to the main RCT, a pragmatic review of the pilot data in terms of assessing treatment efficacy (based on clinical response), tolerability and patient acceptability will be undertaken. STOP/GO guidelines will be used to determine whether to proceed forward to the main RCT, and we propose then to take forward the preferred method of FMT delivery into a randomised double-blind placebo-controlled trial with a clinical efficacy outcome (clinical remission).

The first STOP/GO decision will be on which route of FMT to take forward to the full RCT. The pilot is not powered to show differences in the two modes of delivery (NG or COLON); therefore, a decision or recommendation on which route to use in the main trial requires expert judgement and cannot be made based on a purely numerical process. The following data will be reviewed by the IOC:

1. The proportion of patients who achieve a clinical response following FMT by the NG route (A threshold of achieving a clinical response in around 40% of patients treated may be used. This threshold will be agreed after discussions by the Trial Management Group and the IOC prior to any data analysis being performed).
2. The proportion of patients who achieve a clinical response following FMT by the COLON.
3. Whether FMT by either route is associated with a change in faecal calprotectin.
4. Whether FMT by either route is associated with changes in the colonic microbiome and metabolome.
5. Tolerability and safety for each route.
6. Patient acceptability of FMT by the NG route through the qualitative interviews (including advantages and disadvantages of the NG route).

7. Patient acceptability of FMT by the COLON route through the qualitative interviews (including advantages and disadvantages of the COLON route).

Following review of the above data, whether the IOC can recommend a route to take forward to the main trial, will form the basis of the first STOP/GO decision.

1. IOC unable to recommend one particular route as neither route felt to be satisfactory. DECISION: Do not proceed to main RCT.
2. IOC recommends a route of FMT delivery for use in the main RCT. DECISION: Proceed to STOP/GO decision stage 2 of pilot.

Once the route of FMT delivery has been selected, a second STOP/GO assessing feasibility will be used to determine whether to proceed to the main RCT. This will be based on the following:

1. That the IOC are able to recommend a route.
2. That the recruitment of the 30 patients in the pilot averages 0.7 patients per week in each open site.
3. That 10 of the 15 patients in the route cohort selected for the main study received at least 70% of their intended FMT dose. For example, 70% of the NG route (240 g) is 168 g, which requires the patient to receive at least six infusions (30 g; six out of the eight FMT infusions that is, patient can miss no more than two doses); and 70% of the COLON route (360 g) is 252 g, which requires the patient to receive the first infusion by colonoscopy (150 g) and then at least four out of seven of the enema infusions (30 g) that is, patient can miss no more than three doses.
4. That the IOC have not identified any safety concerns.

Following review of the above data, whether the IOC are able to recommend that a full RCT is feasible will form the basis of the second STOP/GO decision.

1. IOC consider an RCT unfeasible. DECISION: Do not proceed to main RCT.
2. IOC consider an RCT feasible. DECISION: Provide report to funder with recommendation of IOC, in order for a final decision to be made on progression to the full RCT.

## Patient and public involvement

Our protocol was developed in consultation with a patient and public involvement (PPI) group referred to as the Clinical Research Ambassador Group, based at University Hospital NHS Foundation Trust. One of the coapplicants is a patient with IBD and also Chairperson for the West Midlands Group of Crohn's and Colitis UK (CCUK) and she helped develop the proposal. The PPI has strong links with CCUK and this will significantly aid dissemination of findings nationally to patients, relatives and health professionals.

As we are aware that acceptability of FMT in UC has not been extensively studied, we designed the pilot study to investigate patient acceptability of the therapy by regular questionnaires, individual interviews and group discussions. We conducted a survey of our own IBD patients from the outpatient department at University Hospital Birmingham. Of the 74 patients surveyed, the vast majority of patients would accept FMT as a treatment for IBD and over 80% would consider involvement in a trial of FMT in IBD. Qualitative interviews with patients will be conducted at baseline and at the end of the study and these findings will feed into the discussions regarding progression to a full-scale RCT. Patients recruited to the study will not necessarily directly assist with the recruitment of other patients.

## Ethics and dissemination

The trial will be performed in accordance with the recommendations guiding physicians in biomedical research involving human subjects, adopted by the 18th World Medical Association General Assembly, Helsinki, Finland, June 1964, amended at the 48th World Medical Association General Assembly, Somerset West, Republic of South Africa, October 1996.

The trial will be conducted in accordance with the Research Governance Framework for Health and Social Care, the applicable UK Statutory Instruments (which include the Medicines for Human Use Clinical Trials 2004 and subsequent amendments and the General Data Protection Regulation (GDPR) (EU) 2016/679 and Human Tissue Act 2008) and Guidelines for Good Clinical Practice. This trial will be carried out under a Clinical Trial Authorisation in accordance with the Medicines for Human Use Clinical Trials regulations.

## Publication policy

At the end of the study, a meeting will be held to allow discussion of the main results among the collaborators prior to publication. The success of the study will depend entirely on the wholehearted collaboration of a large number of doctors, nurses and others. For this reason, the chief credit for the main results will be given not only to the central supervisory committees and/or organisers, but to all those who have collaborated in the trial.

The findings will be reported at national and international gastroenterology meetings and published in peer-reviewed journals.

**Author affiliations**
[1]University of Birmingham Microbiome Treatment Centre, University of Birmingham, Birmingham, UK
[2]Department of Immunology and Immunotherapy, Medical School, University of Birmingham, Birmingham, UK
[3]Department of Gastroenterology, St Marks Hospital, London, UK
[4]Gastroenterology Unit, Glasgow Royal Infirmary, Glasgow, UK
[5]Department of Gastroenterology, Heart of England NHS Foundation Trust, Birmingham, UK
[6]Institute of Microbiology and Infection, University of Birmingham, Birmingham, UK
[7]Public Health Laboratory Birmingham, Public Health England Midlands and East Region, Birmingham, UK
[8]Clinical Trials Unit, Institute of Applied Health Research, University of Birmingham, Birmingham, UK
[9]Human Nutrition, School of Medicine, Dentistry and Nursing, Glasgow Royal Infirmary, University of Glasgow, Glasgow, UK
[10]Department of Paediatric Gastroenterology, Royal Hospital for Children Glasgow, Glasgow, UK
[11]Institute of Applied Health Research, University of Birmingham, Birmingham, UK

[12] Public Health, University of Birmingham, Birmingham, UK
[13] Warwick Medical School, Microbiology and Infection, University of Warwick, Coventry, UK
[14] Crohn's and Colitis UK, Saint Albans, UK
[15] Department of Gastroenterology, Queen Elizabeth Hospital, Birmingham, UK

**Correction notice**  This article has been corrected since it was published. The first author name has been updated.

**Contributors**  MNQ, MY, JS, CB, NS, PH, VM, ALH, DG, RH, NJI, LM, SL, CH, NC and TI were all involved in developing the clinical protocol and authoring the manuscript, with MNQ being the lead author, TI the chief ivestigator. NJI is the senior statistician on the project. She was involved in the trial design, protocol development and led the development of the proposed statistical analyses. CH is the trial statistician and was involved in protocol development and produced the statistical analysis plan for this study. SL is the trial coordinator and with LM developed the trial management and governance aspects of the manuscript. KG, NJL, CQ contributed the mechanistics and bioinformatics expertise. JM and CM authored the qualitative research section of the manuscript. We thank our patient advisor NC for her help with developing this protocol. We used the SPIRIT checklist when writing this protocol.

**Funding**  This study is funded by the National Institute for Health Research (NIHR) as part of the Efficacy and Mechanism Evaluation (EME) programme.

**Competing interests**  None declared.

**Patient consent for publication**  Not required.

**Ethics approval**  Ethical approval for this study has been obtained from the East Midlands-Nottingham Research Ethics Committee (REC 17/EM/0274).

**Provenance and peer review**  Not commissioned; externally peer reviewed.

**ORCID iDs**
Mohammed Nabil Quraishi http://orcid.org/0000-0003-2338-8397
Nicholas James Loman http://orcid.org/0000-0002-9843-8988
Christel McMullan http://orcid.org/0000-0002-0878-1513

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
