## [Reviewer comments · BMJ Open]

ARTICLE DETAILS

TITLE (PROVISIONAL)	Stop-Colitis pilot trial protocol. A prospective, open-label, randomised pilot study to assess two possible routes of Faecal Microbiota Transplant delivery in patients with ulcerative colitis
AUTHORS	Quraishi, Mohammed Nabil; Yaichin, Mehmet; Blackwell, Clare; Segal, Jonathan; Sharma, Naveen; Hawkey, Peter; McCune, Victoria; Hart, Ailsa L.; Gaya, Daniel; Ives, Natalie; Magill, Laura; Loi, Shrushma; Hewitt, Catherine; Gerasimidis, Konstantinos; Loman, Nicholas; Hansen, Richard; McMullan, Christel; Mathers, Jonathan; Quince, Christopher; Crees, Nicola; Iqbal, Tariq

VERSION 1 – REVIEW

REVIEWER	DEAN HARRIS Dept of Colorectal Surgery Morrison Hospital Swansea SA6 6NL Wales
REVIEW RETURNED	22-Apr-2019

GENERAL COMMENTS	Mechanistic objectives: "Engraftment of donor microbiota in recipients by culturing donor stool to strain level."- why is culturing rather than sequencing (16S or NGS) being used? Which organisms in particular are sought? Later in the manuscript (page 14) no mention of culturing, only sequencing. Page 8 Mechanistic outcome measures: "Measures of microbiome (faecal and mucosal);" should be better described/defined here (rather simplistic) such as diversity indices, etc Donor criteria: please justify use of non-smokers from the literature More detail about the donor sample processing needed: is there any anaerobic protection used? How long will the donor aliquot be stored for in the event of adverse effects? Patient inclusion/exclusion criteria: will incur wide heterogeneity here as patients may range from mild newly diagnosed treatment naïve disease, to years of chronic disease on immunomodulatory therapy. Can this spectrum be accounted for in the analysis? How was the dosing regime decided for each group? What evidence was it based on? Why is there a 4 week gap between NG treatments? There is no placebo arm, please justify Dietary diary being collected for donors, what about recipients, particularly after treatment? Symptom progression during follow up- will patients be withdrawn? How will calprotectin and CRP data be analysed, in comparison to baseline level (proportion/% change), or number that return to normal?
--

	How will AE/SAEs be handled? What AEs are to be expected? Stop/go: how will you proceed if both routes are equivalent? Has the EME funded the subsequent RCT, or will this be a new application?
--	--

REVIEWER	Dr Sam Costello The Queen Elizabeth Hospital Adelaide, Australia
REVIEW RETURNED	26-Apr-2019

GENERAL COMMENTS	This pilot study is thoughtfully designed to answer the question of which delivery method to use in a larger RCT of FMT for UC. I have a few suggested edits Intro: "the importance of Roseburia hominis, Faecalibacterium prausnitzii and Akkermansia muciniphila driving inflammation in UC" This sentence is potentially misleading as these organisms are thought to be protective, ie inversely correlated with inflammation. Page 7: Ideally a primary outcome should be clearly defined, and other outcomes listed as secondary Page 8: Qualitative outcome data. A questionnaire or some questions being posed should be listed in the document. Page 12: Colonoscopic treatment: Loperamide takes 2-3 hours to reach maximum effect and so ideally should be given prior to enema or colonoscopy. Typically patients will lose enema in first 30 min following insertion. 100mls is also a large volume to hold with active UC. 50-60mls may be more manageable. Qualitative data: This is a semi-structured interview. What are the questions being asked? Short chain fatty acid measurement. Total stool weight will need to be measured over a time period to accurately measure SCFAs. Fibre: A distinction should be made between resistant starches and other fibre as resistant starches are particularly important to the physiology of the colonic environment.
---

REVIEWER	Konstantinos Triantafyllou University of Athens
REVIEW RETURNED	19-May-2019

GENERAL COMMENTS	My major concern is the study outcomes measurements. Authors indicate that they will use a composite end-point for these measurements. Although they list the components of this endpoint, they do not provide any score(s) that the outcomes should meet. More over, authors state that "The pilot is not powered to show differences in the two modes of delivery (NG or COLON); therefore a decision or recommendation on which route to use in the main trial requires expert judgement and cannot be made based on a purely numerical process." In conjunction with the absence of formal
---

	sample size estimation, the confusion for the outcomes measurements is culminated. Thus, composite endpoint cut off score(s) should be set and formal sample size estimation should be done for this trial, in my opinion
--	--

VERSION 1 – AUTHOR RESPONSE

Reviewer: 1

1. Mechanistic objectives: “Engraftment of donor microbiota in recipients by culturing donor stool to strain level.”- why is culturing rather than sequencing (16S or NGS) being used? Which organisms in particular are sought? Later in the manuscript (page 14) no mention of culturing, only sequencing.

Thank you for pointing this out. Although culturing was initially planned, we have decided to not do this as part of the pilot. I have therefore taken reference to this out of the paper. Our microbial mechanistics are focused on 16S rRNA sequencing and targeted metagenomics sequencing.

2. Page 8 Mechanistic outcome measures: “Measures of microbiome (faecal and mucosal);” should be better described/defined here (rather simplistic) such as diversity indices, etc

We agree and have now updated the outcome measure to ‘Measures of microbiome (faecal and mucosal); specifically shifts in alpha diversity following FMT.’ Although we will be exploring changes in beta diversity indices and relative abundances in microbial taxa, the sample size of the pilot is not adequate to include this as an outcome measure.

3. Donor criteria: please justify use of non-smokers from the literature. More detail about the donor sample processing needed: is there any anaerobic protection used?

We recognise that the eligibility of smokers for stool donation is a grey area. However there have been quite a few studies now demonstrating that smokers have an altered gut microbiome compared to non-smokers (references below). Consequently, we collectively felt that a more cautious approach would be to exclude smokers especially in view of the unknown long term risks associated with FMT.

- Shanahan, E. R. et al. Influence of cigarette smoking on the human duodenal mucosa-associated microbiota. *Microbiome* 6, 150 (2018).
- Allais, L. et al. Chronic cigarette smoke exposure induces microbial and inflammatory shifts and mucin changes in the murine gut. *Environ. Microbiol.* 18, 1352–1363 (2016).
- Lee, S. H. et al. Association between Cigarette Smoking Status and Composition of Gut Microbiota: Population-Based Cross-Sectional Study. *J Clin Med* 7, (2018).

Thank you for the comment about donor stool preparation. Due to space constraints we were limited on the level of detail to go in to. However, we have now added the lines:

‘Stool collection and processing were performed under aerobic conditions.’

‘Further information on the donor sample processing is in the protocol and a lab SOP for donor preparation (available on request).’

We appreciate that anaerobic collection and processing of stool would be logical as gut bacteria are primarily facultative anaerobes. However, aerobic collection and processing has had no effect in outcomes for use of FMT for recurrent / refractory *Clostridium difficile* infection. There are no head to

head studies exploring superiority of anaerobic FMT preparation of aerobic FMT. We agree that donor FMT preparation optimisation deserves detailed exploration as part of a separate study.

4. How long will the donor aliquot be stored for in the event of adverse effects?

Donor aliquots are stored for 10 years as per BSG guidelines and University of Microbiome Treatment Centre policy.

5. Patient inclusion/exclusion criteria: will incur wide heterogeneity here as patients may range from mild newly diagnosed treatment naïve disease, to years of chronic disease on immunomodulatory therapy. Can this spectrum be accounted for in the analysis?

We appreciate that there will be a wide heterogeneity in patient characteristics in the pilot. The randomisation into this pilot was minimised by smoking status and partial Mayo score to account for differing levels of disease severity at baseline. Due to the size of the pilot, it was not practical to minimise for many variables, but in a full RCT, we would plan to minimise on other variables to ensure balance across treatment groups. The aim of this pilot study was to assess two modes of delivery of FMT which we do not believe will be affected by whether the patient is newly diagnosed or has chronic disease. In fact, recruiting both means that we can assess the safety of FMT delivery in different types of patients. The analysis of the clinical data will be adjusted for the smoking status and Mayo score.

6. How was the dosing regime decided for each group? What evidence was it based on? Why is there a 4 week gap between NG treatments?

Thank you for this important query. There is currently no good evidence base for dosing regimen of FMT for the treatment of UC. From the four RCTs that have been published so far, the studies with a higher number of total FMT treatments are associated with greater response rates. Consequently a total of 8 infusions was a pragmatic choice for the study. For patient convenience, we decided that for the nasogastric route this should be delivered as two 4 consecutive infusions with a 3 week interval between them.

7. There is no placebo arm, please justify

This study is a pilot study to determine which FMT administration route (naso-gastric or colon) should be investigated in a randomised double-blind, placebo-controlled trial. At the end of the pilot, decisions will be made regarding the feasibility of a full-scale randomised double-blind, placebo-controlled trial and, if deemed feasible, which route of administration should be used in such a study. Therefore, we do not believe a placebo arm for this pilot study is necessary. A placebo will be appropriate and is planned for the next stage of the study (the large scale randomised double-blind, placebo-controlled trial).

8. Dietary diary being collected for donors, what about recipients, particularly after treatment?

Dietary diary is collected for both donors and recipients. This has now been amended in the manuscript.

9. Symptom progression during follow up- will patients be withdrawn?

Patients who demonstrate significant worsening of symptoms or Mayo scores will not be withdrawn from the study. They may stop trial treatment if considered necessary, but they will continue to be

followed-up for the duration of the trial, so that an intention-to-treat analysis can be undertaken. Patients will be treated as per standard of care as per local hospital policy for management of UC, and any additional treatments received due to disease progression and/or relapse will be collected.

10. How will calprotectin and CRP data be analysed, in comparison to baseline level (proportion/% change), or number that return to normal?

Calprotectin and CRP will be analysed as a comparison to baseline level. Further details are provided as part of the response to Reviewer 2's comment 6.

11. How will AE/SAEs be handled? What AEs are to be expected?

Within the pilot trial we are collecting both AEs and SAEs. On each of the FMT treatment forms and the 8 and 12 week follow-up forms, information on targeted AEs such as diarrhoea, nausea, abdominal pain etc. are being collected. We also ask for information on any other AEs which the patient may have experienced.

SAEs are being handled in the same way as for any CTIMP trial. Any AE that fulfils the definition of a serious adverse event should be reported on a trial specific serious adverse event form.

12. Stop/go: how will you proceed if both routes are equivalent?

If both routes are found to be equivalent, and it is not possible to distinguish between the two modes of delivery, then discussion would be needed as to whether the full-scale trial should be a three arm trial (nasogastric vs. colonic vs. placebo), or whether the RCT remains as a two arm trial (FMT vs. placebo), but route of FMT delivery can be either nasogastric or colonic.

13. Has the EME funded the subsequent RCT, or will this be a new application?

The subsequent RCT has been funded by EME, however progression to this phase of the study is dependent on the analysis / outcomes of this pilot study.

Reviewer: 2

Reviewer Name: Dr Sam Costello

Institution and Country: The Queen Elizabeth Hospital, Adelaide, Australia Please state any competing interests or state 'None declared': None

1. Intro: "the importance of Roseburia hominis, Faecalibacterium prausnitzii and Akkermansia muciniphila driving inflammation in UC" This sentence is potentially misleading as these organisms are thought to be protective, ie inversely correlated with inflammation.

Thank you for pointing this out. We have now updated this statement.

2. Page 7: Ideally a primary outcome should be clearly defined, and other outcomes listed as secondary

This is a pilot study so does not have defined primary and secondary outcomes as per phase I-IV RCTs. The primary objective of this pilot is to determine which route of FMT delivery (nasogastric or colonic) should be investigated in a large-scale double-blind, placebo-controlled RCT. This will be assessed through both quantitative and qualitative data. For the quantitative clinical data, we have detailed in the paper that clinical response is the primary measure of clinical efficacy for the

STOP/GO, and that the other clinical outcomes are secondary measures of clinical efficacy for the STOP/GO. This follows the CONSORT extension for pilot and feasibility studies reporting standard.

3. Page 8: Qualitative outcome data. A questionnaire or some questions being posed should be listed in the document.

In response to Reviewer 2's comment, a summary of the questions asked to patients relating to the qualitative outcomes was included in page 8. In addition, a copy of the interview schedule is included as a supplementary document.

4. Page 12: Colonoscopic treatment: Loperamide takes 2-3 hours to reach maximum effect and so ideally should be given prior to enema or colonoscopy. Typically patients will lose enema in first 30 min following insertion. 100mls is also a large volume to hold with active UC. 50-60mls may be more manageable.

This is an important point. Randomisation to either route (nasogastric or colon) is performed if partial Mayo score confirms active disease (partial Mayo score of ≥ 4 and ≤ 8). However, evidence of active disease at times may not be as well-defined (if there is a functional overlap). Consequently, we felt that endoscopic confirmation of evidence of colonic mucosal inflammation during the colonoscopy would be needed in such cases. If the participant is randomised to the colonic route of delivery, FMT would be delivered during withdrawal of the colonoscope from the caecum and loperamide given as soon as the patient is back in their room for recovery. Measures taken that include minimising air insufflation, delivery of a larger proportion FMT into proximal colon and reducing mobilisation for an hour post the FMT delivery would reduce the risk of loss of FMT. For subsequent FMT treatments, patients are given loperamide immediately after administration of the enema.

We accept that a lower enema volume may be better managed by the patient, however this would be at the risk of inadequate delivery of FMT into the left colon. Additionally from our experience of lower GI treatment of recurrent and refractory CDI, patients have not had any significant difficulties in managing 100mls of FMT delivered via an enema. Additionally, the amount of faecal suspension given via an enema has varied between 150 and 500 in studies exploring use of FMT for CDI and is recommended in current UK guidelines - Mullish, B. H. et al. The use of faecal microbiota transplant as treatment for recurrent or refractory *Clostridium difficile* infection and other potential indications: joint British Society of Gastroenterology (BSG) and Healthcare Infection Society (HIS) guidelines. *Gut* 67, 1920–1941 (2018).

5. Qualitative data: This is a semi-structured interview. What are the questions being asked?

As there were too many interview questions to include within the manuscript, copies of the interview schedules used for this qualitative study have been added as supplementary documents.

6. Short chain fatty acid measurement. Total stool weight will need to be measured over a time period to accurately measure SCFAs.

Thank you for this important comment. Similar to any metabolite measured in faeces, their concentration is dependent on the total amount produced, absorbed and the total volume of faeces this is diluted. As the reviewer proposes the amount of total output per bowel movement or even per day might be a better estimate of true colonic production. However, we would still be unable to

account for the net production of SCFA in the colon as only 5% of the total amount produced is estimated to be excreted in faeces.

Here, faecal SCFA will be expressed in three different ways; a) per g of 'wet' faeces, b) per g of 'dry' faeces and as c) relative abundance of each SCFA to the total faecal SCFA. We have previously demonstrated that depletion or reduction in dietary fibre corresponded to a significant decrease in the concentrations of SCFA per g of stool and the % representation of each SCFA in healthy people as well as in people with IBD (references below). These observations offer confidence that any changes in the SCFA signal we will observe in per spot stool analysis will reflect luminal SCFA production. Finally, serial or cumulative samples collection would increase further participation burden.

- Gerasimidis, K., Nikolaou, C. K., Edwards, C. A. & McGrogan, P. Serial fecal calprotectin changes in children with Crohn's disease on treatment with exclusive enteral nutrition: associations with disease activity, treatment response, and prediction of a clinical relapse. *J. Clin. Gastroenterol.* 45, 234–239 (2011).
- Gerasimidis, K. et al. Decline in presumptively protective gut bacterial species and metabolites are paradoxically associated with disease improvement in pediatric Crohn's disease during enteral nutrition. *Inflamm. Bowel Dis.* 20, 861–871 (2014).

7. Fibre: A distinction should be made between resistant starches and other fibre as resistant starches are particularly important to the physiology of the colonic environment.

This is a very interesting suggestion but unfortunately the UK food composition tables, similar to other international food composition tables, do not list the amount of resistant starch in food. The latter is also difficult to estimate as it depends on cooking conditions (fresh pasta vs cold pasta), how ripe or not a fruit/vegetable/starchy food is (e.g. ripe banana vs a banana which is not ripe) and GI function and motility which may differ in UC patients compared with healthy subjects.

Reviewer: 3

1. My major concern is the study outcomes measurements. Authors indicate that they will use a composite end-point for these measurements. Although they list the components of this endpoint, they do not provide any score(s) that the outcomes should meet. More over, authors state that "The pilot is not powered to show differences in the two modes of delivery (NG or COLON); therefore a decision or recommendation on which route to use in the main trial requires expert judgement and cannot be made based on a purely numerical process." In conjunction with the absence of formal sample size estimation, the confusion for the outcomes measurements is culminated. Thus, composite endpoint cut off score(s) should be set and formal sample size estimation should be done for this trial, in my opinion

We appreciate the reviewer's comment regarding the study outcome measures. This is a pilot study assessing a novel treatment (FMT) in a complex disease. It is therefore appropriate that any decision to progress to a full trial is based on a mixture of both quantitative and qualitative data. For the key quantitative outcomes (clinical response and adherence), we have stated progression guidelines within the STOP-GO with cut-offs (see pages 16-17 of the paper). For qualitative data it is not possible to set cut-off scores, but this data will provide important information on the acceptability of the two modes of delivery and must form part of the decision process. The STOP-GO process was discussed at length and agreed with the funder NIHR EME.

VERSION 2 – REVIEW

REVIEWER	Dr Sam Costello The Queen Elizabeth Hospital South Australia Australia
REVIEW RETURNED	07-Jul-2019

GENERAL COMMENTS	All of the previous concerns that I had have been addressed
---

REVIEWER	Associate Professor Konstantinos Triantafyllou National and Kapodistrian University of Athens, Greece
REVIEW RETURNED	12-Jul-2019

GENERAL COMMENTS	Authors have successfully addressed my concern
--